# The role of drop shape in impact and splash

Qingzhe Liu [1], Jack Hau Yung Lo [1✉], Ye Li[2], Yuan Liu [1], Jinyu Zhao[1] & Lei Xu [1✉]

The impact and splash of liquid drops on solid substrates are ubiquitous in many important fields. However, previous studies have mainly focused on spherical drops while the non-spherical situations, such as raindrops, charged drops, oscillating drops, and drops affected by electromagnetic field, remain largely unexplored. Using ferrofluid, we realize various drop shapes and illustrate the fundamental role of shape in impact and splash. Experiments show that different drop shapes produce large variations in spreading dynamics, splash onset, and splash amount. However, underlying all these variations we discover universal mechanisms across various drop shapes: the impact dynamics is governed by the superellipse model, the splash onset is triggered by the Kelvin-Helmholtz instability, and the amount of splash is determined by the energy dissipation before liquid taking off. Our study generalizes the drop impact research beyond the spherical geometry, and reveals the potential of using drop shape to control impact and splash.

[1] Department of Physics, The Chinese University of Hong Kong, Hong Kong, China. [2] CAS Key Laboratory of Quantitative Engineering Biology, Shenzhen Institute of Synthetic Biology, Shenzhen Institutes of Advanced Technology, Chinese Academy of Sciences, Shenzhen, China. ✉email: hylo@cuhk.edu.hk; xuleixu@cuhk.edu.hk

The impact of a liquid drop on a solid substrate is a common phenomenon, which plays a significant role in many important fields, such as agriculture, printing, surface coating, spray cooling, and transmission of respiratory diseases. Drop impact happens in a blink of an eye. With bare eyes, we can only see the final outcomes: splash, deposit, or bounce. Splash typically happens when impact speed is high. With the advance in imaging technology, profound understanding has been developed through studying the "slow-motion" video of the drop impact process[1,2], which is conceptually divided into different stages for different underlying mechanisms, while many interesting phenomena and important mechanisms have just been discovered recently[3–14].

While extensive studies have been performed on the spherical drops[1–14], much less is understood on the non-spherical counterparts, which frequently appear in many actual situations. For example, raindrops are deformed by aerodynamic stress and have flattened bottoms[15,16], charged drops develop sharp tips when they approach oppositely charged substrates or when the charge amount is close to the Rayleigh limit[17–19], oscillating drops exhibit a variety of non-spherical shapes[20–22], and the external electric or magnetic field may also change the shape of drops[23–27]. As a result, our knowledge is largely limited to one specific drop geometry, and the general picture across different shapes remains missing.

In this work, we illustrate the fundamental role of drop shape in impact and splash by experimentally realizing various drop shapes. Experiments show that different drop shapes produce large differences in spreading dynamics, splash onset, and splash amount. However, underlying all these differences we discover universal mechanisms which are valid across various drop shapes. Our study generalizes the basic understanding of drop impact beyond the spherical geometry, and brings a fundamental breakthrough in this research field. It also reveals the potential of using drop shape to control impact and splash for practical applications, which is distinct from the conventional parameters like impact velocity, air pressure[28–32], material properties of liquid[33,34], and substrate[35–39].

## Results and discussion

Due to the surface tension of liquid, drops are typically spherical in experiments and it is difficult to probe shape's fundamental influence on impact and splash. Using ferrofluid drops and accurately controlled magnetic field, we systematically generate various drop shapes and tackle this fundamental issue. As shown in Fig. 1a, a free-falling ferrofluid drop first passes through a magnetic coil, which generates a strong magnetic field and stretches the drop into a long spindle-like shape. The magnetic field is then quickly turned off before drop impact, and the drop starts to oscillate under surface tension. The timing of turning off the magnetic field is precisely controlled by a laser trigger and an off-delay timer. By carefully adjusting the turn-off time of the magnetic field, we can achieve various drop shapes at the moment of impact. Figure 1b shows examples of drop cross-sectional shapes, in 3D they are axisymmetric around the vertical central axis. Their impact dynamics are illustrated by high-speed movies in Supplementary Movies 1–6. The solid substrates are smooth and dry microscope glass slides (see "Methods": "Materials and setup"). We have also repeated the experiments on polymethyl methacrylate (PMMA or acrylic glass) surface and piranha-cleaned glass, whose surface energies are different from the microscope slides, and obtained the same results (see Supplementary Note 5)[40–42], demonstrating the general validity of our results.

Great effort is taken to make sure that the magnetic field and drop oscillation do not affect the impact process: the magnetic field reduces to negligible level before the impact (smaller than 0.30 mT or 1.3% of the original value, see Supplementary Note 1), and the oscillation period (~31 ms) is much longer than the impact process (smaller than 0.76 ms) such that the drop shape is stable throughout the impact process (see Supplementary Note 2).

When a drop impacts onto a solid substrate with high enough speed, it typically goes through the three essential stages: (1) spreading rapidly along the substrate, (2) taking off from the surface to create the onset of splash, and (3) breaking into satellite droplets and splashing. To obtain a deep and thorough understanding, we study shape's influence on all three stages and illustrate them one by one.

We first demonstrate the overall effect of shape by comparing the impacts of three drops with distinct shapes in Fig. 1c (also see Supplementary Movie 7): the first three columns respectively illustrate the impact of an elongated, a flattened, and a spherical drop from the side view. Apparently, the spreading dynamics and the splash outcome vary significantly with shape: the elongated drop spreads the slowest but produces most splatters while the flattened one spreads the fastest but generates least satellite droplets, and the spherical drop is in between. The fourth column presents the bottom view of the spherical case (i.e., the third column). From the bottom view, we can accurately determine the spreading dynamics in stage-1, and the onset of splash in stage-2. As indicated by the yellow arrow, the bright interference pattern due to liquid sheet flying off the substrate clearly manifests the onset of splash. In Fig. 1d we show the splatter patterns left by the ferrofluid stains, from which we can quantitatively measure the amount of splash for stage-3. Clearly, our experiment enables a quantitative study of shape's influence on all three stages.

**Spreading dynamics governed by the superellipse model in stage-1.** We first illustrate the influence of drop shape on the spreading dynamics in stage-1. Numerous studies have shown that spherical drops follow a simple one-half power law in this early stage spreading[43–48]: $r = 2\sqrt{RVt}$, where $r$ is the radial position of contact line, $t$ is the time after contact, $V$ is the impact velocity, and $R$ is the drop radius. The dimensionless format is:

$$r' = 2t'^{1/2} \tag{1}$$

Here $r' \equiv r/R$ and $t' \equiv tV/R$ are dimensionless radial position and time. Note that some other models give the same 1/2 power law with slightly different numerical prefactors[49–51]. Despite its simplicity, Eq. (1) has been generally verified in many studies[43–48]. Here we also test it with our spherical drops: Fig. 2a shows our spherical drop data for various impact velocities and liquids, including the ferrofluid, and our data agree excellently with the black line of Eq. (1).

For non-spherical drops, however, the scaling law $r = 2\sqrt{RVt}$ fails. First and obviously, the radius $R$ is ill-defined for non-spherical drops. Second and more importantly, our measurements show that the fundamental exponent of the power law can significantly deviate from 1/2, as shown in Fig. 2c. This fundamental deviation in the exponent cannot be solved by looking for an effective radius. In fact, we now show that the exponent depends on the geometry of the drop, and the 1/2 power only holds for the special case of spheroidal geometry. Similarly, the dimensionless prefactor of 2 in Eq. (1) also changes when the drop geometry varies.

Here we construct a model that generalizes the existing law of Eq. (1) to non-spherical geometry. In this model, the drop shape is described by the equation of a superellipse:

$$\left|\frac{x}{a}\right|^n + \left|\frac{y}{b}\right|^n = 1 \tag{2}$$

When $n = 2$, it is an ellipse and the spherical shape is the most special case of $a = b$. Based on $n$, we divide different shapes into

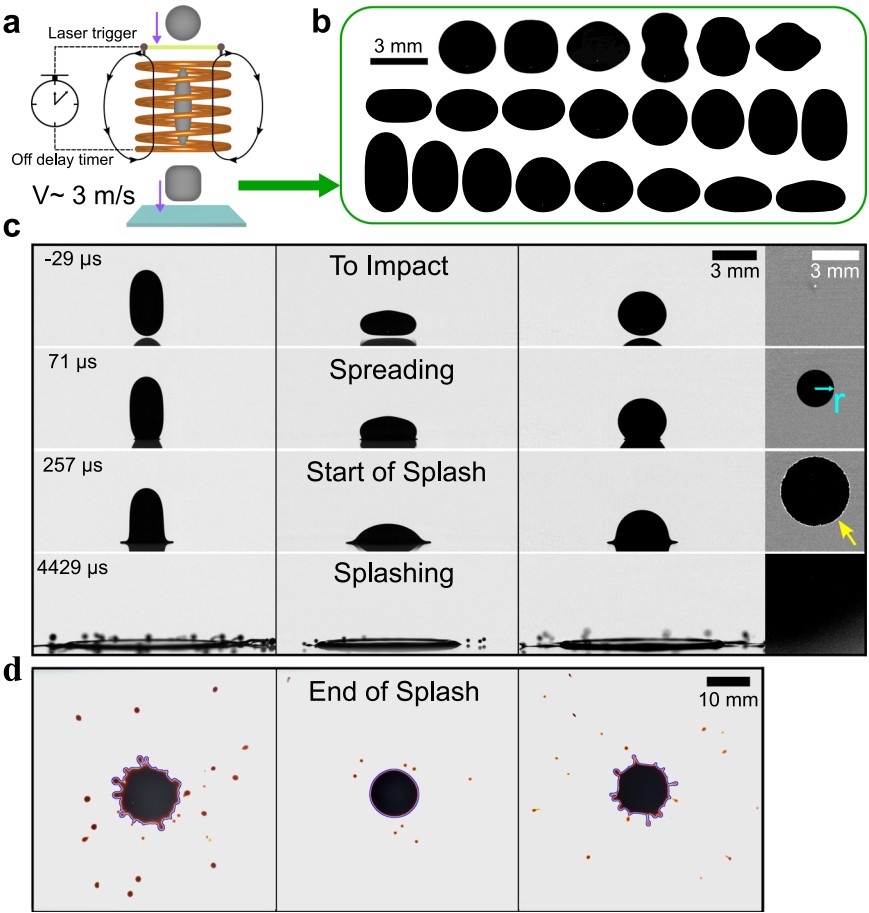

**Fig. 1 Experimental setup and typical impact events by different shaped drops. a** A schematics of our setup (not in scale). A free-falling ferrofluid drop first passes through a magnetic coil, which stretches the drop into a long spindle-like shape. The magnetic field is then turned off and the falling drop starts to oscillate across many different shapes due to surface tension. The timing of turning off the magnetic field is precisely controlled by a laser trigger and an off-delay timer. By fine-tuning the turn-off time of magnetic field, we realize different drop shapes at the impact moment. **b** Examples of drop cross-sectional shapes, in 3D they are axisymmetric around the vertical central axis. **c** Side-view snapshots of impact events by three typical shapes: elongated (column 1), flattened (column 2), and spherical (column 3) drops. They have the same impact velocity, $V = 2.9 \pm 0.2$ m/s. Column 4 shows the bottom view of column 3. The cyan arrow indicates the radial position of contact line, $r$. The yellow arrow indicates the bright interference pattern due to the liquid sheet flying off the substrate, whose first appearance indicates the onset of splash. **d** Corresponding splatter patterns of the three drops in (**c**), which indicate the amount of splash. The blue curve shows the border between the parent drop and the satellite droplets.

three categories: $n = 2$ is defined as "round" for its similarity to a sphere, $n > 2$ is defined as "flat" due to its more flattened bottom than a sphere, and $n < 2$ is defined as "sharp" because of its more sharpened bottom than a sphere. The schematics in Fig. 2b illustrates these definitions straightforwardly. The superellipse can describe most of our shapes, as shown in Figs. 1b and 2c, except a few rare examples with concave interfaces. In our model, the superellipse exponent, $n$, represents a fundamental feature of shape. We thus define $n$ as the sharpness (or flatness) of a drop, which will be shown to determine the power law exponent of the spreading dynamics. Besides $n$, the other essential quantities in Eq. (2) are the horizontal and vertical length scales, $a$ and $b$, which will also be shown to affect the spreading dynamics. By introducing the superellipse description of drop shape into the conventional volume conservation model (see Supplementary Note 3 for derivation), we obtain a general shape-dependent expression for the early spreading dynamics:

$$t' = 1 - {}_2F_1\left(\frac{-1}{n}, \frac{2}{n}; \frac{n+2}{n}; r'^n\right) \quad (3)$$

Here ${}_2F_1$ is the hypergeometric function, $t' \equiv tV/b$ is the dimensionless time and $r' \equiv r/a$ is the dimensionless radial

contact line position. By neglecting higher order terms, Eq. (3) can be simplified into a power law:

$$r' \approx \left(\frac{n(n+2)}{2}\right)^{1/n} t'^{1/n} \quad (4)$$

Comparing with the spherical result, $r' = 2t'^{1/2}$, our expression is similar but more general: the 1/2-power changes into $1/n$ and the prefactor 2 also changes into an $n$-dependent quantity. Besides depending on $n$, $r' \equiv r/a$ and $t' \equiv tV/b$ also depend on $a$ and $b$. Thus, by knowing all the shape quantities, $n$, $a$ and $b$, we obtain a parameter-free and universal model for various shapes' impact dynamics. We further illustrate this model with two special cases below.

For a sphere, $n = 2$ and $a = b = R$, Eq. (4) reduces to the classical result, $r = 2\sqrt{RVt}$, as we naturally expect. For a spheroidal drop, i.e., an ellipsoid of revolution, $n = 2$ but $a \neq b$, Eq. (4) reduces to $r = 2\sqrt{(a^2/b)Vt}$. Comparing it with $r = 2\sqrt{RVt}$, we find that by defining an effective radius, $R = a^2/b$, the classical equation is still valid for a spheroid. Note that $a^2/b$ is exactly the local radius of curvature at the spheroid's bottom. This treatment can serve as a very useful correction for

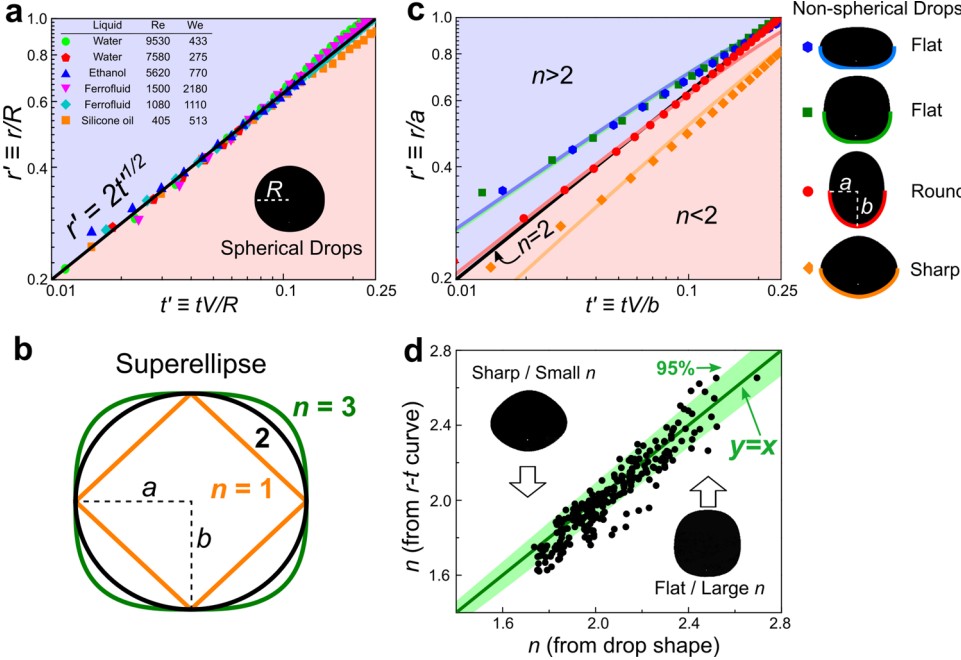

**Fig. 2 The spreading dynamics explained by a superellipse model. a** Dimensionless spreading radius versus time for spherical drops. The experimental data agree excellently with the theoretical scaling law, $r' = 2t'^{1/2}$ (the thick black line), for various liquids (including the ferrofluid) and impact velocities. **b** The superellipse model. We construct a general model by fitting a non-spherical shape to a superellipse: $n = 2$ is defined as 'round', $n < 2$ is defined as 'sharp', and $n > 2$ is defined as 'flat'. $n$ is defined as the sharpness, $a$ and $b$ are the semi-major and semi-minor axes. **c** Verifying the superellipse model with four non-spherical drop shapes. The colored symbols are experimental data for different shapes, and the colored curves are predictions from the superellipse model, Eq. (3), without any fitting parameter. Four examples are shown: two $n > 2$ or 'flat' examples are on the top, one $n = 2$ or "round" example is in the middle, and one $n < 2$ or "sharp" example is at the bottom. The two "flat" examples have distinct outlooks: one is disc-like and the other is square-shaped (see the images at right). However, they exhibit almost identical spreading dynamics due to their similar sharpness $n$ (blue and green symbols). The right panel shows the drop images and the colored curves at the bottom are the superellipse fittings. **d** A comprehensive test on our model with many drop shapes. $X$-axis is the $n$ values directly measured from the drop shape before impact (Eq. (2)). $Y$-axis is the $n$ values obtained by fitting spreading dynamics to the superellipse model (Eq. (3)) after impact. Two sets of data are mostly within 95% confidence interval, which demonstrates an excellent agreement.

many real situations in which spherical drops distort into spheroids[22,52–55].

The predictions of our model are in good agreement with the experimental data. To illustrate, we plot the theoretical predictions (solid curves) together with the experimental data in Fig. 2c: note that no fitting parameter is used and all the parameters, $n$, $a$, $b$, and $V$, are obtained from the high-speed images before impact shown in the right panel. With our model, we can quantitatively predict the spreading dynamics after the impact by only knowing the shape information and velocity before the impact.

More interestingly, Eq. (4) also predicts that the dimensionless form of spreading dynamics only depends on the drop sharpness (or flatness) $n$. As a result, a disc-like drop and a square-shaped drop (see the images in Fig. 2c), which have very distinct outlooks and aspect ratios but similar $n$, exhibit almost identical spreading dynamics. The overlap of green and blue curves in Fig. 2c quantitatively demonstrates this result. Therefore, the sharpness (or flatness), defined by $n$, is a fundamental feature that determines the dimensionless spreading dynamics just by itself, and different $n$ values provide a universal description for various drop shapes.

Besides the four typical examples shown in Fig. 2c, we further make a comprehensive test across much more shapes, as shown in Fig. 2d. On one hand, we systematically change the shapes of drops, and directly measure $n$ from these shapes before impact. On the other hand, we measure the $r(t)$ curve after impact and fit it with Eq. (3), obtain $n$ from the spreading dynamics. These two sets of $n$ are then compared as $x$ and $y$-axis values in Fig. 2d, and

an excellent agreement is observed. These extensive tests unambiguously verify the universal validity of our superellipse model for various shapes.

**Splash onset triggered by the Kelvin–Helmholtz instability in stage-2.** After rapid spreading, the liquid sheet flies off the surface and reaches the second stage: the onset of splash. Now let us study the shape's influence on splash onset. As shown in Fig. 1c column four, we can accurately identify the splash onset by the bright interference pattern (indicated by the yellow arrow). As a result, the essential quantities at the onset, such as the onset location ($R_{onset}$), onset time ($t_{onset}$), and onset spreading velocity ($v_{onset}$), can all get precisely measured so that we can probe shape's influence on them in details.

Figure 3a demonstrates three snapshots at time $t_{onset}$ for three typical shapes at the same impact velocity, $V = 2.9 \pm 0.2$ m/s. Clearly, the onset location $R_{onset}$ varies significantly with the drop vertical length $L$: the larger $L$ is, the smaller $R_{onset}$ will be. We further systematically measure the dependence of $R_{onset}$ on the dimensionless drop length $L/D$ ($D = 2.91$ mm being the diameter of an equivalent sphere with the same volume), as plotted it in Fig. 3b: a dramatic change is observed with $R_{onset}$ decreases exponentially with $L/D$. Similarly, the onset time $t_{onset}$ also varies significantly with drop shape, as shown in Fig. 3b inset. Apparently, drop shape induces dramatic variations in the splash onset.

However, a very different result appears for the onset velocity, $v_{onset}$. As shown in Fig. 3c, although the three curves of spreading

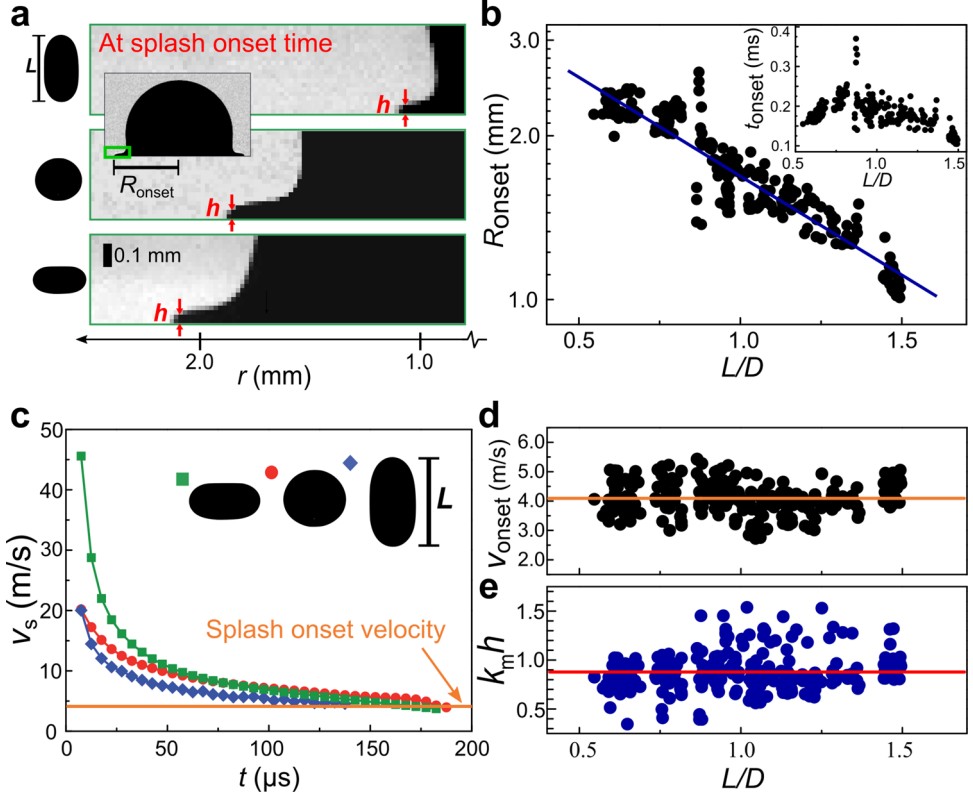

**Fig. 3 The onset of splash explained by the Kelvin–Helmholtz instability criterion. a** Zoomed-in snapshots at the moment of splash onset for three typical shapes at the same impact velocity, $V = 2.9 \pm 0.2$ m/s. Clearly a longer drop with a larger $L$ has a smaller splash onset location $R_{onset}$. The red arrows indicate how we measure the liquid sheet thickness, $h$. The inset shows a zoomed-out image. **b** The splash onset location, $R_{onset}$, versus the dimensionless drop length, $L/D$, with $D$ the diameter of an equivalent sphere. An exponential relation, $R_{onset} \propto \exp(-0.37\,L/D)$, appears. Inset: The splash onset time $t_{onset}$ also varies significantly with shape. **c** Spreading velocity versus time, $v_s(t)$, for the three typical shapes shown in (**a**). Although the three curves initially differ significantly, their last data points, $v_{onset}$, are very close. **d** The splash onset velocities, $v_{onset}$, for various shapes. They stay close to an average value, $4.1 \pm 0.5$ m/s, indicated by the horizontal line. **e** The splash onset data for various shapes agree well with the general criterion: $k_m h \sim 1$. Here $k_m$ is the wave number of the fastest growing mode of Kelvin–Helmholtz (KH) instability in the Knudsen regime[56]. The horizontal line indicates the mean of our data, 0.9. Some large deviations may come from the limited spatial resolution of our camera in the $h$ measurements (see Supplementary Fig. 7).

velocity are very different initially, their last data points, $v_{onset}$, are close to each other, as indicated by the horizontal line. Extensive tests for much more shapes verify the robustness of this agreement, as shown in Fig. 3d, and the large variations in their initial spreading velocities are given in Supplementary Fig. 6.

A constant splash onset velocity implies a general mechanism underlying the splash onset. For a spherical drop, our previous study has revealed a splash onset criterion: $k_m h \sim 1$, with $h$ the liquid sheet thickness at the edge and $k_m$ the wave number of the fastest growing mode of Kelvin–Helmholtz (KH) instability in the Knudsen regime[56]. This criterion has been experimentally verified for spherical drops of various liquids and impact velocities[56]. We now generalize it to non-spherical geometries and test whether this criterion is universally valid for different shapes.

By measuring the thickness $h$ and the spreading speed $v_{onset}$ at the moment of splash, we can experimentally determine $k_m h$ for various drop shapes, as plotted in Fig. 3e. Without any fitting parameter, all data points stay close to a constant value of $0.9 \pm 0.2$, which confirms a universal splash criterion, $k_m h \sim 1$, for various shapes. This criterion also theoretically explains the rather close $v_{onset}$ values: $v_{onset} \propto k_m$ and the $k_m$ values are rather close at the onset time (see Supplementary Fig. 7), which leads to close $v_{onset}$ values. To summarize, despite the broad variations in $R_{onset}$ and $t_{onset}$ induced by shape change, we reveal a general splash onset criterion, $k_m h \sim 1$, which gives a universal description for splash onset across different shapes.

**Splash amount determined by energy dissipation in stage-3.** After splash onset, the satellite droplets are continuously produced, and the impact enters the final splash stage. By studying shape's influence in this stage, we once again find significant variations in splash amount, which however can be explained by energy dissipation. We have already illustrated how the splash outcomes vary significantly with shape by comparing the photos of splatters in Fig. 1d. Here we quantify the amount of splash by measuring the total area of stains from splatters (see Method: Satellite droplets collection) and make an extensive comparison for various shapes. In all experiments, we fix the drop volume (13 μL) and impact speed ($2.9 \pm 0.2$ m/s), while change the drop shape as the only variable. The total stain area, $A$, is plotted against the dimensionless length, $L/D$, in Fig. 4a. We observe an exponential dependence: $A \propto \exp(1.10\,L/D)$, where the volumetric equivalent diameter, $D$, becomes the characteristic length. As the drop length $L$ changes from $0.5D$ to $1.5D$, the total stain area $A$ increases about one order of magnitude! Such a significant variation suggests shape as a powerful control over splash.

To explain this surprising phenomenon, we construct an energy dissipation model. It is reasonable to assume that the amount of splash should be inversely dependent on the energy dissipated during spreading, $E_{dis}$. The more energy is dissipated, the less energy is left to produce splash. Therefore, we may assume a simple inverse proportional relation, $A \propto 1/E_{dis}$, between the splash amount and the energy dissipation. The

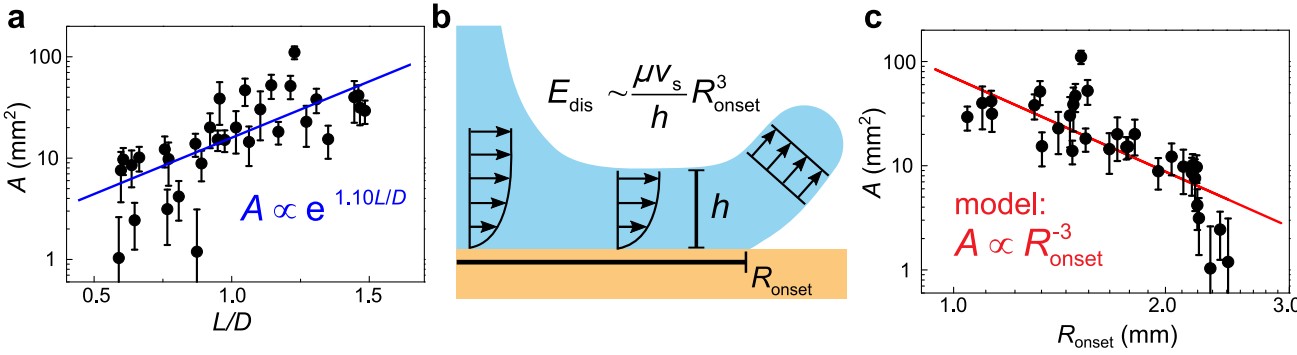

**Fig. 4 The amount of splash explained by an energy dissipation model. a** The total area of splatter stains, $A$, versus the dimensionless drop length, $L/D$, in log-linear scale. $D$ is the diameter of an equivalent sphere with the same volume. The error bars represent standard deviations. We find an exponential increase of splash amount with respect to the dimensionless length, $A \propto \exp(1.10 \, L/D)$. **b** Schematics showing the energy $E_{\mathrm{dis}}$ dissipated in the spreading liquid sheet due to strong viscous shear before taking off. Here $\mu$ is the liquid viscosity, $v_{\mathrm{s}}$ is the spreading velocity, $h$ is the liquid film thickness and $R_{\mathrm{onset}}$ is the onset radius of splash. **c** Total area of splatter stains, $A$, versus the splash onset location, $R_{\mathrm{onset}}$, in log-log scale. The error bars represent standard deviations. The data agree with the power law, $A \propto R_{\mathrm{onset}}^{-3}$, predicted by the energy dissipation model.

viscous dissipation is estimated as $E_{\mathrm{dis}} \sim (\mu v_{\mathrm{s}}/h)R_{\mathrm{onset}}^3$ (see Supplementary Note 4 for derivation), where $\mu$ is the liquid viscosity, $v_{\mathrm{s}}$ is the spreading velocity, $h$ is the liquid film thickness, which is approximately the boundary layer thickness[57], and $R_{\mathrm{onset}}$ is the onset radius of splash. Similar approximations can successfully deduce the scaling of maximum spreading diameter of drops by energy conservation[58–62]. In our model only the region $r < R_{\mathrm{onset}}$ contributes to viscous dissipation, because the liquid sheet would detach from the substrate at the position $R_{\mathrm{onset}}$ at timescale $R_{\mathrm{onset}}/v_{\mathrm{s}}$; the liquid sheet which is detached from the substrate has much smaller shear rate and hence neglectable energy dissipation (see Fig. 4b). Therefore, our model predicts that: $A \propto 1/E_{\mathrm{dis}} \propto R_{\mathrm{onset}}^{-3}$. Because different drop shapes produce large variations in $A$ and $R_{\mathrm{onset}}$ without any change in liquid properties, it gives a good opportunity to test this prediction across a broad range. As shown in Fig. 4c, the power law of $A \propto R_{\mathrm{onset}}^{-3}$ (the solid line) is indeed observed, consistent with our model's prediction. By correlating the amount of splash $A$ with the energy dissipation $E_{\mathrm{dis}}$, our model provides a general understanding on the splash production by various shaped drops.

We note that the model deviates from the rightmost data points, whose impact conditions are very close to the threshold of splashing. Near this threshold, the ejected satellite droplets have very small kinetic energy and land in close proximity to the parent drop, and then get "swallowed" by the spreading parent drop, as shown in Supplementary Fig. 9. This leads to an underestimation of the splash amount and the deviation. We also note that the peanut shape drop (see Fig.1b) produces the largest amount of splash. This shape exhibits a highly concave interface near the neck region, which can be considered as two droplets connected together by the neck, and two impacts by two consecutive droplets may probably produce more splatters than one big drop.

We also note that air pressure variation may cause change in splashing amount[28,29]. However, because for our daily life and industrial applications the atmospheric pressure is the typical and most relevant condition, our experiments' high reproducibility demonstrates its accuracy under this most important condition.

To summarize, by systematically varying the drop shape, we study its fundamental effect on drop impact in three essential stages. In each stage, different shapes produce dramatic variations. However, underlying all these variations we discover general mechanisms, which are universally valid across different shapes. Our study expands the basic knowledge of drop impact beyond the spherical geometry and reveals the potential of using

drop shape to control impact and splash. This approach has the unique advantage of least modification to the system: it only requires change in drop shape, while keeps all essential system quantities, such as the impact velocity, the liquid properties of drops (for example the surface tension or viscosity) and the surface properties of substrates (for example, the contact angle or roughness) all unchanged.

## Methods

**Materials and setup.** The ferrofluid is purchased from FerroTec (model EFH1), which is oil-based and paramagnetic[27], with the viscosity 8 cP, density $1.2 \times 10^3$ kg/m$^3$ and surface tension 19 mN/m (at 20 °C, no magnetic field). The viscosity is measured by a rheometer (MCR 301, Anton-Paar), the surface tension is measured by the pendant drop method. The substrates are 1 mm thick glass microscope slides (Lab'IN Co., HK). The glass slides are washed by acetone, IPA, and deionized water in supersonic bath before experiment. The contact angle between glass slides and water drops is 26°, the literature value of the surface energy of the glass slides is 68 mN/m[41]. The ferrofluid, whose surface tension is 19 mN/m, wets the glass slide. The drops are produced from a needle with a syringe pump (TJP-3A, Longer), released from pre-determined heights. The impact process is recorded by two synchronized high-speed cameras (SA-Z, Photron) at the frame rate up to 200,000 fps, from both the bottom and the side. All experiments are conducted in ambient environment at the room temperature 20 °C.

**Critical condition for liquid sheet take off.** As shown in Fig. 3e, the splash onset velocities $v_{\mathrm{onset}}$ of the expanding liquid sheet agree with the critical condition derived from the Kelvin–Helmholtz instability in Knudsen regime in our previous study:[56] $k_m h \sim 1$. Here $h$ is the thickness of the liquid sheet at the edge, as shown in Fig. 3a, $k_m = \sqrt{2/(9\pi\gamma)}\rho_a c v_s/\sigma$ is the wave number of the fastest growing mode, $\gamma = 1.4$ is the adiabatic gas constant, $\rho_a$ is the density of air, $c$ is the speed of sound in air, $v_s$ is the velocity of liquid front, $\sigma$ is the surface tension of the liquid.

**Satellite droplets collection.** To collect all the satellite droplets produced by splash, the impact substrate is surrounded by white paper as illustrated in Fig. S8. The satellite droplets are collected by either the substrate or the paper walls and leave black stains, which are photographed by a digital camera. Then we measure the total stain area from both the substrate and the paper walls, which quantifies the total amount of splash. Note that the front and back side has no paper wall because we need a window for high-speed photography. To account for their absence, the stain area collected by the papers on the left and right sides are multiplied by two. Each experiment has typically repeated eight times and the statical average and standard deviation are presented.

## Data availability

The data that support the findings of this study are available from the corresponding author upon reasonable request.

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

## Acknowledgements

Funding: L.X. acknowledges the financial support from Hong Kong RGC GRF 14306920, GRF 14306518, CRF C1018-17G, CUHK United College Lee Hysan Foundation Research Grant and Endowment Fund Research Grant and CUHK direct grant 4053354.

## Author contributions

L.X. conceived the project, Q.L., Y.L., J.H.Y.L., and L.X. designed the experiments, Q.L. carried out the experiments, J.H.Y.L. and L.X. constructed the models, J.H.Y.L. and L.X. supervised the research, Q.L., J.H.Y.L., and L.X. prepared the manuscript, and all authors contributed to the data analysis.

## Competing interests

The authors declare no competing interests.
