## [Peer Review File · Nature Communications]

Reviewers' Comments:

Reviewer #1:

Remarks to the Author:

- 1) justify the sentence from line 57-60 with quantities.
- 2) if ellipsoid drop shapes naturally occur, provide evidence or references to show they are not simply a special case seen in ferrofluids. Are they only common when created by nozzles or similar injection systems or do they spontaneously occur?
- 3) Figure 3d and 3e: So $v_{\text{onset}}=4\text{m/s} \pm 50\%$? $k_{\text{mh}}=0.9 \pm 50\%$? (I beg to differ from the assertion that the spread is 0.2 as stated on line 178). Justify why the scatter in these two figures is acceptable.
- 4) Figure 4 a and c both have 5 outliers on the low side and one to the high side. Are these the same data? Do cases which poorly follow the proposed trendline fall into a subcategory with any parametric classification? Explain points whose error bars do not fall upon the trend line.
- 5) The quantitative relationship between drop shape and time span for turning-off/on of the magnetic field is missing. Line# 53-54 - "By carefully adjusting the turn-off time of the magnetic field, we can achieve various droplet shapes at the moment of impact." quantitative information must support this statement. Also, the relationship between the turning off time and two consecutive drop shapes can be established.
- 6) It is claimed that the magnetic field vanishes before the impact (Line#58), interestingly it is further claimed that impact dynamics are free from magnetic field. With this assumption, The drop must regain its original spherical shape but before the impact it is not; please justify and clarify the assumption with the appropriate scaling analysis.
- 7) Line # 63, authors states that "When a droplet impacts onto a solid substrate, it typically goes through three essential stages: (1) spreading rapidly along the substrate, (2) taking off from the surface to create the onset of splash, and (3) breaking into satellite droplets and splashing." This is an incomplete representation since bouncing is observed irrespective of surface energy of the substrates.
- 8) What is the type of ferrofluid used in this study? Is it a water based or oil based ferrofluid. Based on the carrier liquid (diamagnetism or paramagnetism) the drop can have different shapes under magnetic field. Please refer, JCIS, Volume 532, Pages 309-and Langmuir 2016, 32, 30, 7639-7646" for further details.
- 9) A very important detail on the range of magnetic field strength is missing, and discussion on dependency between the strength and drop shape. Moreover, the discussion on the change in interfacial tension and viscosity, due to the presence of the magnetic field, is paramount to comment on the drop shape. Details on the surface tension measurements of the ferrofluid are missing, is it a pendant drop (did you get the shape factor correctly) or Wilhelmy plate (what was the equilibration period)?
- 10) The authors have mentioned the surface tension remains unchanged throughout the study. It is not correct: the surface tension of the magnetic fluid changes not only due to the inclusion of colloidal particles but the applied magnetic field also alters it.
- 11) In order to generalise the conclusion, only glass substrate results are not sufficient, at least 3-4 substrates of widely varying surface energy are necessary to conclusively make a generalized statement.

12) Please justify the drop volume of 15 micro lit;d it is closer to the capillary length scale, in fact if you consider the correct density it is beyond the capillary length scale? Also can you elaborate the negligible gravitational body force with scaling analysis.

13) It seems the viscous dissipation model used in this study is based on boundary layer approximation (Capillary effects during droplet impact on a solid surface Physics of Fluids 8, 650 (1996); <https://doi.org/10.1063/1.868850>), which is suitable for higher contact angle scenario. However, in this study a glass substrate is used which is hydrophilic in nature. It is to be noted that, for lower contact angle the lubrication approximation model, proposed by De Gennes, has been used for viscous dissipation (Dynamics of partial wetting, PG De Gennes). Please provide the details (probably in the supplementary material) of viscous dissipation energy expression used in the line# 203. Also, Reference 45-47 are not sufficient to get this expression hence supplementary material details are necessary.

14) What was the critical height for the droplet splashing in this study, i.e., the distance between the tip of the needle and substrate?

Brian Fleck, University of Alberta

Reviewer #2:

Remarks to the Author:

Comments to Both Author and Editor:

Dear Editor,

Thank you for inviting me to review the article, "The role of droplet in impact and splash". The manuscript of Liu et al., reports on the effect of droplet shapes on impact and splash of the droplet. The authors show that the sharpness of the droplet would determine the dimensionless spreading dynamics just by itself. A new model is also proposed for understanding the splash production by various shaped droplets. Though this work is interesting and is relevant to the journal domain, but comprehensive consideration, this manuscript could be published after a major revision. Some specific comments are below:

1. In the discussion section on page 6, the author said: "it only requires a change in droplet shape, while keeps all essential system quantities such as the impact velocity, the liquid properties of droplets...and the surface properties of substrates... all unchanged". In my opinion, the contents of the results are not enough to support this conclusion. For example, only the glass is used in the experiment, and its surface property is not clear, is it hydrophobic or hydrophilic surface? How about the spreading behavior of droplets with different shapes on other types of substrate? More related experiments should be done.

2. Figure 3 indicated that droplet shape induces dramatic variations in the splash onset conditions. The author chose three typical shapes to test, however, I found the centers of mass of three droplets are at different heights, so do the authors have considered the difference of energy among the three situations?

3. Xu et al. found that ambient pressure is also an important factor to influence the splashing behavior of droplets, and when it is less than a critical value, the splashing will be suppressed. So the ambient pressure will influence the accuracy of the conclusion concluded in the manuscript?

Reply to Reviewer #1

1) justify the sentence from line 57-60 with quantities.

Reply: We thank the reviewer for pointing out the ambiguous sentences. Following this suggestion, we have revised line 57-60 with quantities as follows:

“Great effort is taken to make sure that the magnetic field and droplet oscillation do not affect the impact process: the magnetic field reduces to negligible level before the impact ($<0.30\text{mT}$ or $<1.3\%$ of the original value, see SI), and the oscillation time ($\sim 31\text{ms}$) is much longer than the impact process ($<0.76\text{ms}$) due to the high impact velocity ($\sim 3\text{m/s}$) such that the droplet shape is stable throughout the impact process (see SI)”.

More details of quantitative justifications are given below:

To ensure that the magnetic field becomes negligible before droplet impact, we measure the magnetic field versus time at the impact point with a Hall sensor, as shown below. The measurement shows that when the magnetic field is on, the field strength is 23.2mT , and due to inductance the response time of turning off is about 15ms . Therefore, we turn off the magnetic field at least 20ms before the droplet impact happens, which guarantees that the magnetic field strength is negligible ($<0.3\text{mT}$ or $<1.3\%$ of the original value) during impact. Further measurements show that the liquid properties such as the surface tension and the viscosity remain unchanged under such a weak magnetic field, and thus the influence from magnetic field is negligible (see Table S1 and S2 below).

Fig. S1 Magnetic field strength at the impact position versus time.

Table S1 Viscosity of the ferrofluid measured by rheometer under different magnetic field strength. The minimum field strength in the measurement region increases from 0 to 3.00 mT, which is 10 times stronger than the field during our impact process (<0.3 mT). However, no change in the viscosity is observed. Therefore, the weak residue of magnetic field in our experiment does not induce any observable change in viscosity.

Minimum magnetic field in the region (mT)	Maximum magnetic field in the region (mT)	Viscosity (mPa·s)
0	0	7.70 ± 0.01
0.50	0.77	7.69 ± 0.01
2.03	2.19	7.70 ± 0.01
3.00	3.25	7.70 ± 0.01

Table S2 Surface tension of the ferrofluid measured by pedant drop method under different magnetic field strength. The minimum field strength in the measurement region increases from 0 to 2.83 mT, which is over 9 times stronger than the field during our impact process (<0.3 mT). However, no change in the surface tension is observed. Therefore, the weak residue of magnetic field in our experiment does not induce any observable change in surface tension.

Minimum magnetic field in the region (mT)	Maximum magnetic field in the region (mT)	Surface Tension (mN/m)
0	0	18 ± 2
0.11	0.13	18 ± 2
2.02	2.39	18 ± 2
2.83	3.34	17 ± 2

To demonstrate that drop oscillation does not influence the impact, we show the following calculation. The timescale of droplet oscillation is $\tau = \pi\sqrt{\rho R^3/2\sigma} \approx 31$ ms calculated by Rayleigh oscillation frequency, where ρ is the liquid density, σ is the surface tension, and R is the droplet radius. The characteristic time scale of the impact is defined as $T=L/2V$ with V the impact speed (2.9 ± 0.2 m/s), and L the droplet length. In our study, L varies between 1.59 mm and 4.35 mm, and thus T ranges from 0.28 ms to 0.76 ms, which is much shorter than the oscillation period of 31ms. Therefore, the droplet shape can be considered as stable throughout the entire impact process. We thank the referee for this great question and we have included the above new contents in the Methods and Supplementary Information.

2)if ellipsoid drop shapes naturally occur, provide evidence or references to show they are not simply a special case seen in ferrofluids. Are they only common when created by nozzles or similar injection systems or do they spontaneously occur?

We have listed several common examples of non-spherical drops with references in the 1st paragraph, the examples include raindrops [15,16], charged droplets [17-19], oscillating drops

[20-22], and drops that are affected by external electric field [23, 24] and magnetic field [25-27]. We also added 5 references of recent studies as examples of ellipsoid drops in the main text sentence “This treatment can serve as a very useful correction for many real situations in which spherical droplets distort into spheroids [22,51-54]” (page 3, second to the last paragraph, last sentence). As shown in these examples, non-spherical drop shapes can either occur spontaneously or be created by injection systems, and can occur for both ferrofluids and non-ferrofluids.

The cited references are listed as follows:

15. Beard, K. V. & Chuang, C. A New Model for the Equilibrium Shape of Raindrops. *J. Atmospheric Sci.* **44**, 1509–1524 (1987).
16. Kostinski, A. B. & Shaw, R. A. Droplet dynamics: Raindrops large and small. *Nat. Phys.* **5**, 624–625 (2009).
17. Duft, D., Achtzehn, T., Müller, R., Huber, B. A. & Leisner, T. Rayleigh jets from levitated microdroplets. *Nature* **421**, 128 (2003).
18. Ristenpart, W. D., Bird, J. C., Belmonte, A., Dollar, F. & Stone, H. A. Non-coalescence of oppositely charged drops. *Nature* **461**, 377–380 (2009).
19. Beroz, J., Hart, A. J. & Bush, J. W. M. Stability Limit of Electrified Droplets. *Phys. Rev. Lett.* **122**, 244501 (2019).
20. Beard, K. V., Ochs, H. T. & Kubesh, R. J. Natural oscillations of small raindrops. *Nature* **342**, 408 (1989).
21. Becker, E., Hiller, W. J. & Kowalewski, T. A. Experimental and theoretical investigation of large-amplitude oscillations of liquid droplets. *J. Fluid Mech.* **231**, 189–210 (1991).
22. Thoraval, M.-J., Takehara, K., Etoh, T. G. & Thoroddsen, S. T. Drop impact entrapment of bubble rings. *J. Fluid Mech.* **724**, 234–258 (2013).
23. Vlahovska, P. M. Electrohydrodynamics of Drops and Vesicles. *Annu. Rev. Fluid Mech.* **51**, 305–330 (2019).
24. Yun, S. & Lim, G. Ellipsoidal drop impact on a solid surface for rebound suppression. *J. Fluid Mech.* **752**, 266–281 (2014).
25. Afkhami, S. *et al.* Deformation of a hydrophobic ferrofluid droplet suspended in a viscous medium under uniform magnetic fields. *J. Fluid Mech.* **663**, 358–384 (2010).
26. Timonen, J. V. I., Latikka, M., Leibler, L., Ras, R. H. A. & Ikkala, O. Switchable Static and Dynamic Self-Assembly of Magnetic Droplets on Superhydrophobic Surfaces. *Science* **341**, 253–257 (2013).
27. Ahmed, A., Qureshi, A. J., Fleck, B. A. & Waghmare, P. R. Effects of magnetic field on the spreading dynamics of an impinging ferrofluid droplet. *J. Colloid Interface Sci.* **532**, 309–320 (2018).
51. Li, E. Q., Thoraval, M.-J., Marston, J. O. & Thoroddsen, S. T. Early azimuthal instability during drop impact. *J. Fluid Mech.* **848**, 821–835 (2018).

52. Wischniewski, C. & Kierfeld, J. Spheroidal and conical shapes of ferrofluid-filled capsules in magnetic fields. *Phys. Rev. Fluids* **3**, 043603 (2018).
53. Ahmed, A., Fleck, B. A. & Waghmare, P. R. Maximum spreading of a ferrofluid droplet under the effect of magnetic field. *Phys. Fluids* **30**, 077102 (2018).
54. Yun, S. & Kim, I. Spreading Dynamics and the Residence Time of Ellipsoidal Drops on a Solid Surface. *Langmuir* **35**, 13062–13069 (2019).

3) Figure 3d and 3e: So $v_{\text{onset}}=4\text{m/s} \pm 50\%$? $k_{\text{mh}}=0.9 \pm 50\%$? (I beg to differ from the assertion that the spread is 0.2 as stated on line 178). Justify why the scatter in these two figures is acceptable.

Due to the overlap of data points, the actual spread of the data is much smaller than they look like on the figures. There are 256 data points in both Figures 3d and 3e, and many of them are overlapping near the mean value. As a result, the outliers look more pronounced than they actually are. To clarify the ambiguity, we added two histograms in Fig.S11 in the Supplementary Information as shown below. According to these data, the mean and standard deviation of v_{onset} is: $4\text{m/s} \pm 0.5\text{m/s}$ (or $\pm 13\%$), and the mean and standard deviation of k_{m^*h} is: 0.9 ± 0.2 (or $\pm 22\%$). Both are much less than $\pm 50\%$ as suspected by the referee. Therefore, our original statement in line 178 is correct.

In addition, we also made a minor correction to Fig.3d: we found that the three uppermost outliers in Fig.3d are produced by the errors of the edge detection computer program and corrected this error. These three data points are now closer to the mean value and we apologize for this mistake!

Fig. S5 Histograms of v_{onset} (left) and k_{m^*h} (right) corresponding to Fig3d and 3e of the main text. The total data points are $N=256$. The mean and the standard deviation of v_{onset} are 4.1 ms and 0.5 ms, and the mean and the standard deviation of k_{m^*h} are 0.9 and 0.2.

4) Figure 4 a and c both have 5 outliers on the low side and one to the high side. Are these the same data? Do cases which poorly follow the proposed trendline fall into a subcategory with any parametric classification? Explain points whose error bars do not fall upon the trend line.

Reply: The referee is correct, the 5 outliers in Fig. 4a and c are indeed the same data points. We explain two distinct types of deviation from the trendline as the following. For the 4 outliers significantly below the trendline in Fig. 4c, they are at the impact conditions very close to the threshold of splashing. Near this threshold, the ejected satellite droplets have very small kinetic energy and sometimes fail to detach from the parent drop. Even when they do detach from the parent drop, they often land in close proximity to the parent drop, and get “swallowed” by the spreading parent drop, as shown in Fig.S9 below. This leads to an underestimation of the splash amount and thus the 4 outliers are below the trendline.

Fig. S9 Ejected droplets get “swallowed” by the spreading mother droplet. a, Ejected droplets land in close proximity to the parent droplet. The image is captured soon after impact ($\sim 2\text{ms}$) by high speed camera. **b,** When we measure the total stain area, those ejected droplets have already been “swallowed” by the spreading parent droplet. The image is captured at time $\sim 8\text{s}$ after impact. The red box indicates the same region shown in panel (a). **c,** The magnified image of the region inside the red box, which is the same region as (a).

For the 1 outlier significantly above the trendline in Fig.4c, it corresponds to the unusual peanut-shape droplet, as shown in Fig.1b. This shape exhibits a highly concave interface near the neck region, which can be considered as two droplets connected together by the neck. As a result, its impact is similar to two impacts by two consecutive droplets. Two impacts by two consecutive droplets tend to produce more splatters than one big droplet. This is because the first droplet wets the surface and produces a small liquid puddle for the second one to impact upon, which usually produces more splatters than one single impact on a dry surface. As a result, the peanut shape droplet produces more splash due to its similarity as two consecutive droplets.

We thank the referee for this careful question and we have added one new paragraph to explain these deviations in the current manuscript (page 5, second to the last paragraph).

5) The quantitative relationship between drop shape and time span for turning-off/on of the magnetic field is missing. Line# 53-54 - “By carefully adjusting the turn-off time of the magnetic field, we can achieve various droplet shapes at the moment of impact.” quantitative information must support this statement. Also, the relationship between the turning off time and two consecutive drop shapes can be established.

Reply: We thank the referee for this great suggestion. As suggested by the referee, we provide a gallery of droplet shapes with turn-off time as shown below. The number below each shape is the time interval in millisecond between the moment of droplet detected by a laser trigger and the moment of turning off the magnetic field. By tuning this turn-off time, various shapes at the impact moment are produced, as shown in the gallery. This figure is now included in the Supplementary Information (Fig. S2).

Fig. S2 Drop shapes with turn-off time of magnetic field in millisecond

6) It is claimed that the magnetic field vanishes before the impact (Line#58), interestingly it is further claimed that impact dynamics are free from magnetic field. With this assumption, the drop must regain its original spherical shape but before the impact it is not; please justify and clarify the assumption with the appropriate scaling analysis.

Reply: We thank the referee for this great question. Our previous claim of field vanish is indeed inaccurate and we have revised it to “the magnetic field reduces to negligible level before the impact (<0.30mT or <1.3% of the original value, see Methods)”. Under such a low magnetic field, the ferrofluid properties such as the surface tension and the viscosity remain the same as zero magnetic field. Therefore, we can safely claim that the field has negligible influence on the impact process.

The droplet does not regain spherical shape because it takes time for the droplet to relax back to the spherical shape, while the impact happens before that. After the magnetic field is turned off,

the droplet starts to oscillate under surface tension. The period of droplet oscillation is $\tau = \pi\sqrt{\rho R^3/2\sigma} \approx 31\text{ms}$ calculated by the Rayleigh oscillation frequency, with ρ the liquid density, σ the surface tension, and R the droplet radius. It takes more than five oscillation periods or more than 150ms for the droplet to return to the spherical shape, while the impact occurs between 21ms to 65ms after the field is turned off. As a result, the droplet does not regain its spherical shape at the moment of impact but exhibits various non-spherical shapes during the oscillation. We thank the referee for this great question and we have included all these new information into the Supplementary Information.

7) Line # 63, authors states that “When a droplet impacts onto a solid substrate, it typically goes through three essential stages: (1) spreading rapidly along the substrate, (2) taking off from the surface to create the onset of splash, and (3) breaking into satellite droplets and splashing.” This is an incomplete representation since bouncing is observed irrespective of surface energy of the substrates.

Reply: The referee is correct. Indeed, the droplet may bounce off the substrate when the impact velocity is low and we thank the referee for pointing it out. We have modified this inaccurate statement as the following (changes are underlined):

“When a droplet impacts onto a solid substrate with high enough speed, it typically goes through three essential stages: (1) spreading rapidly along the substrate, (2) taking off from the surface to create the onset of splash, and (3) breaking into satellite droplets and splashing.”

8) What is the type of ferrofluid used in this study? Is it a water based or oil based ferrofluid. Based on the carrier liquid (diamagnetism or paramagnetism) the drop can have different shapes under magnetic field. Please refer, JCIS, Volume 532, Pages 309-and Langmuir 2016, 32, 30, 7639–7646” for further details.

The ferrofluid used in this study is purchased from FerroTec (model EFH1), which is oil-based and paramagnetic, identical to the previous literature [Ahmed, *Journal of Colloid and Interface Science* **532**, 309–320 (2018)]. We have added this information and corresponding literature to the revised manuscript (see Materials and setup in Methods) and we thank the referee for this question.

9) A very important detail on the range of magnetic field strength is missing, and discussion on dependency between the strength and drop shape. Moreover, the discussion on the change in interfacial tension and viscosity, due to the presence of the magnetic field, is paramount to comment on the drop shape. Details on the surface tension measurements of the ferrofluid are missing, is it a pendant drop (did you get the shape factor correctly) or Wilhelmy plate (what was the equilibration period)?

Reply: We thank the referee for this question. We clarify that the different droplet shapes are not realized by applying different magnetic field strengths. Instead, we always apply a constant magnetic field of 23mT to elongate the droplet, and then turn off the field to make the droplet to oscillate under surface tension. By choosing different moment to turn off the field, different droplet shapes are realized at the moment of impact. The magnetic field reduces to a negligible level before the impact (as shown below), and the weak residue field ($<0.3\text{mT}$) induces no observable effect on viscosity and surface tension.

Fig. S1 Magnetic field strength at the impact position versus time.

Table S1 below shows the viscosity measured by a rheometer at different magnetic field strengths. There is no observable change in viscosity even at 3.0 mT, which is ten times larger than our experimental condition ($<0.3\text{mT}$).

Similarly, Table S2 below shows the surface tension measured by the pendant drop method at different magnetic field strengths, and there is no observable change as well.

Table S1 Viscosity of the ferrofluid measured by rheometer under different magnetic field strength. The minimum field strength in the measurement region increases from 0 to 3.00 mT, which is 10 times stronger than the field during our impact process ($<0.3\text{ mT}$). However, no change in the viscosity is observed. Therefore, the weak residue of magnetic field in our experiment does not induce any observable change in viscosity.

Minimum magnetic field in the region (mT)	Maximum magnetic field in the region (mT)	Viscosity (mPa·s)
0	0	7.70 ± 0.01
0.50	0.77	7.69 ± 0.01
2.03	2.19	7.70 ± 0.01
3.00	3.25	7.70 ± 0.01

Table S2 Surface tension of the ferrofluid measured by pedant drop method under different magnetic field strength. The minimum field strength in the measurement region increases from 0 to 2.83 mT, which is over 9 times stronger than the field during our impact process (<0.3 mT). However, no change in the surface tension is observed. Therefore, the weak residue of magnetic field in our experiment does not induce any observable change in surface tension.

Minimum magnetic field in the region (mT)	Maximum magnetic field in the region (mT)	Surface Tension (mN/m)
0	0	18 ± 2
0.11	0.13	18 ± 2
2.02	2.39	18 ± 2
2.83	3.34	17 ± 2

To summarize, we did not control the drop shape with magnetic field strength but use the turn off time to control it. The field decays to a very weak magnitude at the impact moment, and under such a weak field, both viscosity and surface tension do not exhibit any change. We measure the surface tension with pedant drop method. We thank the referee for these valuable questions.

10) The authors have mentioned the surface tension remains unchanged throughout the study. It is not correct: the surface tension of the magnetic fluid changes not only due to the inclusion of colloidal particles but the applied magnetic field also alters it.

Reply: As mentioned in the last question, because the magnetic field at the impact time is weak (<0.3mT), we do not observe any change in the surface tension under such a weak field. We understand that the surface tension may change under a strong enough magnetic field, which is probably the scenario the referee is concerned about. However, such a scenario does not occur in our experiment and thus it is irrelevant to our study.

11) In order to generalize the conclusion, only glass substrate results are not sufficient, at least 3-4 substrates of widely varying surface energy are necessary to conclusively make a generalized statement.

Reply: We thank the referee for this great question. Following this suggestion, we performed our experiments on three different substrates. The first substrate, which is identical to that of the main text, is a glass microscope slide washed by acetone, IPA, and deionized water. The other two substrates are polymethyl methacrylate (PMMA or acrylic glass) surface and piranha-cleaned glass, which have lower and higher surface energy than the microscope slide respectively. The contact angles between a water droplet and the acrylic glass, microscope slide, and piranha-cleaned glass are $74\pm 4^\circ$, $26\pm 4^\circ$, and $7\pm 3^\circ$ respectively. The corresponding literature values of the surface energy of the acrylic glass, microscope slide, and piranha-cleaned glass are 42 mN/m, 68 mN/m and 83

mN/m respectively [Kowalski, *J. Coat. Technol. Res.* **10**, 879–885 (2013); Rhee, *J. Mater. Sci.* **12**, 823–824 (1977)]. The ferrofluid, whose surface tension is 19 mN/m, wets all three substrates.

The data of microscope slide (Black) is overlaid with the new data of acrylic glass (Pink) and piranha-cleaned glass (Green) as shown below, and they overlap nicely. Therefore, our conclusion holds for substrates with different surface energy. This is also consistent with the previous literature that substrate wetting property has little or no effect on high-speed impact dynamics and splash [Latka, A., Boelens, A. M. P., Nagel, S. R. & de Pablo, J. J. Drop splashing is independent of substrate wetting. *Phys. Fluids* **30**, 022105 (2018)].

Fig. S10 The data of microscope slide (Black) are overlaid with the new data of acrylic glass (Pink) and piranha-cleaned glass (Green). The three sets of data overlap with each other. (a)-(c), the

droplet spreading dynamics of three shapes on the three substrates overlap with each other. The main panels show the spreading velocity versus time and the insets show the spreading radius versus time. (d), the splash onset location (main panel) and onset time (inset) on the three substrates agree with each other. (e), splash onset velocities on three substrates overlap with each other. (f), the splash amount on the three substrates overlap with each other. (g), the splash amount on the three substrates overlap with each other and they all agree with the model.

12) Please justify the drop volume of 15 micro liter, it is closer to the capillary length scale, in fact if you consider the correct density it is beyond the capillary length scale? Also can you elaborate the negligible gravitational body force with scaling analysis.

Reply: As the referee has pointed out, the droplet radius (1.46mm) is indeed comparable and slightly larger than the capillary length (1.27mm). For high-speed droplet impact and splashing, the inertia dominates over the gravitational body force. We quantify their ratio with the Froude number, $V_0/\sqrt{g \cdot R}$, which compares the inertia with the gravity effect (here V_0 is the impact velocity, g is gravitational acceleration, and R is the drop radius). In our experiment, the Froude number is about 25, which is significantly larger than 1 and indicates the dominant role of the inertia effect and the negligible role of gravity. Therefore, the gravitational body force can be neglected and the related new information has been included in the Supplementary Information.

13) It seems the viscous dissipation model used in this study is based on boundary layer approximation (Capillary effects during droplet impact on a solid surface *Physics of Fluids* 8, 650 (1996); <https://doi.org/10.1063/1.868850>), which is suitable for higher contact angle scenario. However, in this study a glass substrate is used which is hydrophilic in nature. It is to be noted that, for lower contact angle the lubrication approximation model, proposed by De Gennes, has been used for viscous dissipation (*Dynamics of partial wetting*, PG De Gennes). Please provide the details (probably in the supplementary material) of viscous dissipation energy expression used in the line# 203. Also, Reference 45-47 are not sufficient to get this expression hence supplementary material details are necessary.

Reply: Yes, our viscous dissipation model is indeed similar to the one mentioned by the referee (Pasandideh-Fard *et al*, *Physics of Fluids*, 8, 650, 1996), which is suitable for high contact angle scenario. The reason is the following: for our high-speed impact event, the dynamic contact angle during the fast spreading ($\sim 10\text{m/s}$) is quite large, as shown in Fig.3a. Therefore, the model proposed by Pasandideh-Fard *et al* (*Physics of Fluids*, 8, 650, 1996) is more suitable while the low contact angle model proposed by de Gennes is not suitable for our situation.

Following the referee's suggestion, we have added the detailed expression of the viscous dissipation energy in Method as shown below. The treatment is similar to Pasandideh-Fard *et al* (*Physics of Fluids*, 8, 650, 1996) mentioned by the referee, except two modifications according to

our actual experimental situation: 1) R_{\max} in the literature is replaced by R_{onset} in our experiment, and 2) the timescale is replaced by our splash onset timescale, R_{onset}/v_s .

The viscous dissipation energy of an axisymmetric flow is given by $E_{\text{dis}} = 2\mu \iint \phi dV dt$, where the viscous dissipation function $\phi = \left(\frac{\partial u_r}{\partial r}\right)^2 + \left(\frac{u_r}{r}\right)^2 + \left(\frac{\partial u_z}{\partial z}\right)^2 + \frac{1}{2}\left(\frac{\partial u_r}{\partial z} + \frac{\partial u_z}{\partial r}\right)^2 \approx \frac{1}{2}\left(\frac{\partial u_r}{\partial z}\right)^2$ because of the thin thickness of the spreading liquid sheet. It is further approximated as $E_{\text{dis}} \approx \mu \left(\frac{v_r}{h}\right)^2 \times V \times T$ where μ is the viscosity, v_r is the radial velocity, h is the thickness of liquid sheet, which is approximately the boundary layer thickness [56], V is the sheet volume and T is the timescale. Similar approximations can successfully deduce the scaling of maximum spreading diameter of drops by energy conservation [57-61]. In our case we take $V = \pi h R_{\text{onset}}^2$, $T = R_{\text{onset}}/v_s$, and $v_r = v_s$ where v_s is the spreading velocity of liquid sheet, h is the liquid film thickness, and R_{onset} is the onset radius of splash. Therefore, we get $E_{\text{dis}} \sim \frac{\mu v_s}{h} R_{\text{onset}}^3$.

Fig. 4b Schematics showing the energy E_{dis} dissipated in the spreading liquid sheet due to strong viscous shear before taking off. Here μ is the liquid viscosity, v_s is the spreading velocity, h is the liquid film thickness, and R_{onset} is the onset radius of splash.

Following the referee's suggestion, we have also added more references as shown below:

56. Schroll, R. D., Josserand, C., Zaleski, S. & Zhang, W. W. Impact of a Viscous Liquid Drop. *Phys. Rev. Lett.* **104**, 034504 (2010).
57. Chandra S. & Avedisian C. T. On the collision of a droplet with a solid surface. *Proceedings of the Royal Society of London. Series A: Mathematical and Physical Sciences* **432**, 13–41 (1991).
58. Pasandideh-Fard, M., Qiao, Y. M., Chandra, S. & Mostaghimi, J. Capillary effects during droplet impact on a solid surface. *Physics of Fluids* **8**, 650–659 (1996).

59. Madejski, J. Solidification of droplets on a cold surface. *International Journal of Heat and Mass Transfer* **19**, 1009–1013 (1976).
60. Clanet, C., Béguin, C., Richard, D. & Quéré, D. Maximal deformation of an impacting drop. *Journal of Fluid Mechanics* **517**, 199–208 (2004).
61. Li, X.-H., Zhang, X.-X. & Chen, M. Estimation of viscous dissipation in nanodroplet impact and spreading. *Physics of Fluids* **27**, 052007 (2015).

14) What was the critical height for the droplet splashing in this study, i.e., the distance between the tip of the needle and substrate?

The critical height for the splashing of our *spherical* droplets is about 105 mm (velocity: 1.4 m/s). For comparison, the release heights of our experiments are 365mm - 550 mm (velocity: 2.9m/s – 3.3m/s), which is larger than the critical height of splashing.

To summarize, we thank the referee for the great questions and suggestions. Following these suggestions, we have verified the unchanged viscosity and surface tension of the ferrofluid under our magnetic field strength, performed experiments on different substrates, illustrated the dissipation model in more details, and included extra references and information required by the referee. We believe that our manuscript has improved significantly and we thank the referee for these great suggestions! We hope that the referee can agree that our manuscript is now acceptable.

Reply to Reviewer #2

“Thank you for inviting me to review the article, “The role of droplet in impact and splash”. The manuscript of Liu et al., reports on the effect of droplet shapes on impact and splash of the droplet. The authors show that the sharpness of the droplet would determine the dimensionless spreading dynamics just by itself. A new model is also proposed for understanding the splash production by various shaped droplets. Though this work is interesting and is relevant to the journal domain, but comprehensive consideration, this manuscript could be published after a major revision. Some specific comments are below:”

Reply: we thank the referee for the positive comments that “this manuscript could be published after a major revision”. We address the referee’s specific comments one by one below.

1. In the discussion section on page 6, the author said: “it only requires a change in droplet shape, while keeps all essential system quantities such as the impact velocity, the liquid properties of droplets...and the surface properties of substrates... all unchanged”. In my opinion, the contents of the results are not enough to support this conclusion. For example, only the glass is used in the experiment, and its surface property is not clear, is it hydrophobic or hydrophilic surface? How about the spreading behavior of droplets with different shapes on other types of substrate? More related experiments should be done.

Reply: We thank the referee for this great question. As suggested by the referee, we have performed more experiments on different substrates in the revised manuscript as shown below. We have also added descriptions of the surface properties of these different substrates.

We performed our experiments on three different substrates. The first substrate, which is identical to that of the main text, is a glass microscope slide washed by acetone, IPA, and deionized water. The other two substrates are polymethyl methacrylate (PMMA or acrylic glass) surface and piranha-cleaned glass, which have lower and higher surface energy than the microscope slide respectively. The contact angles between a water droplet and the acrylic glass, microscope slide, and piranha-cleaned glass are $74\pm 4^\circ$, $26\pm 4^\circ$, and $7\pm 3^\circ$ respectively. The corresponding literature values of the surface energy of the acrylic glass, microscope slide, and piranha-cleaned glass are 42 mN/m, 68 mN/m and 83 mN/m respectively [Kowalski, *J. Coat. Technol. Res.* **10**, 879–885 (2013); Rhee, *J. Mater. Sci.* **12**, 823–824 (1977)]. The ferrofluid, whose surface tension is 19 mN/m, wets all three substrates.

The data of microscope slide (Black) is overlaid with the new data of acrylic glass (Pink) and piranha-cleaned glass (Green) as shown below, and they largely overlap. Therefore, our conclusion holds for substrates with different surface energy. This is also consistent with the previous literature that substrate wetting property has little or no effect on high-speed impact dynamics and splash [Latka, A., Boelens, A. M. P., Nagel, S. R. & de Pablo, J. J. Drop splashing is independent of substrate wetting. *Phys. Fluids* **30**, 022105 (2018)].

Fig. S10 The data of microscope slide (Black) are overlaid with the new data of acrylic glass (Pink) and piranha-cleaned glass (Green). The three sets of data overlap with each other nicely. (a)-(c), the droplet spreading dynamics of three shapes on the three substrates overlap with each other. The main panels show the spreading velocity versus time and the insets show the spreading radius versus time. (d), the splash onset location (main panel) and onset time (inset) on the three substrates overlap with each other. (e), splash onset velocities on three substrates overlap with each other. (f), the splash amount on the three substrates overlap with each other. (g), the splash amount on the three substrates overlap with each other and they all agree with the model.

2. Figure 3 indicated that droplet shape induces dramatic variations in the splash onset conditions. The author chose three typical shapes to test, however, I found the centers of mass of three droplets are at different heights, so do the authors have considered the difference of energy among the three situations?

Reply: We thank the referee for this question. We clarify that the difference in centers of mass is negligible compared to the total kinetic energy, because the release height of the drops (>365mm) is over 100 times larger than the diameter of the drops (3mm for spherical drops), and the change in center of mass due to shape is less than the drop diameter and thus is negligible.

3. Xu et al. found that ambient pressure is also an important factor to influence the splashing behavior of droplets, and when it is less than a critical value, the splashing will be suppressed. So the ambient pressure will influence the accuracy of the conclusion concluded in the manuscript?

Reply: We thank the referee for this insightful question. Due to the experimental difficulty, we cannot put our entire magnetic setup into a vacuum chamber and directly test the suppression of splash at lower pressures. However, we believe that our conclusion is quite accurate under atmospheric pressure: despite the typical fluctuation of atmospheric pressure around 500Pa, our results from many different experiments performed at different time are consistent with each other. As shown in the histogram below, we have performed 256 measurements for Figures 3d and 3e, and most data points are overlapping near the mean value. Because for our daily life and industrial applications the atmospheric pressure is the typical and most relevant condition, our experiments' high reproducibility demonstrates its accuracy under this most important condition.

Fig. S5 Histograms of v_{onset} (left) and $k_m \cdot h$ (right) corresponding to Fig3d and 3e of the main text. The total data points are $N=256$. The mean and the standard deviation of v_{onset} are 4.1 ms and 0.5 ms, and the mean and the standard deviation of $k_m \cdot h$ are 0.9 and 0.2.

To summarize, we thank the referee for the great questions and suggestions. Following these suggestions, we have performed new experiments on three different substrates, clarified the negligible influence of change in centers of mass, and verified the accuracy of our results under atmospheric pressure. We believe that our manuscript has improved significantly after these revisions and we thank the referee for these valuable suggestions! We hope that the referee can now agree that our manuscript is acceptable.

Reviewers' Comments:

Reviewer #1:

Remarks to the Author:

I am satisfied with the corrections.

Reviewer #2:

Remarks to the Author:

Globally, the authors answered satisfactorily the questions, but I also have a little suggestion. In responses the authors said that "Because for our daily life and industrial applications the atmospheric pressure is the typical and most relevant condition, our experiments' high reproducibility demonstrates its accuracy under this most important condition", I suggest that the author can clearly state this in the conclusion of the manuscript. After this, I think the paper can be accepted.